# Phytochemical Characterization of *Olea europaea* L. Cultivars of Cilento National Park (South Italy) through NMR-Based Metabolomics

**DOI:** 10.3390/molecules26133845

**Published:** 2021-06-24

**Authors:** Assunta Esposito, Pietro Filippo De Luca, Vittoria Graziani, Brigida D’Abrosca, Antonio Fiorentino, Monica Scognamiglio

**Affiliations:** Dipartimento di Scienze e Tecnologie Ambientali, Biologiche e Farmaceutiche, Università degli Studi della Campania “Luigi Vanvitelli”, via Vivaldi 43, 81100 Caserta, Italy; assunta.esposito@unicampania.it (A.E.); agronomodeluca@alice.it (P.F.D.L.); v.graziani@qmul.ac.uk (V.G.); brigida.dabrosca@unicampania.it (B.D.)

**Keywords:** *Olea europaea*, autochthonous cultivars, biodiversity, NMR, metabolomics, phytochemicals

## Abstract

*Olea europaea* germplasm is constituted by a huge number of cultivars, each one characterized by specific features. In this context, endemic cultivars evolved for a very long period in a precise local area, developing very specific traits. These characteristics include the production and accumulation of phytochemicals, many of which are also responsible for the nutraceutical value of the drupes and of the oils therefrom. With the aim of obtaining information on the phytochemical profile of drupes of autochthonous cultivars of Cilento, Vallo di Diano and Alburni National Park, a metabolomics-based study was carried out on 19 selected cultivars. Multivariate data analysis of ^1^H-NMR data and 2D NMR analyses allowed the rapid identification of metabolites that were qualitatively and/or quantitatively varying among the cultivars. This study allowed to identify the cultivars Racioppella, Guglia, Pizzulella, Oliva amara, and Racioppa as the richest in health-promoting phenolic compounds. Furthermore, it showed a significant variability among the different cultivars, suggesting the possibility of using metabolic fingerprinting approaches for cultivar differentiation, once that further studies aimed at assessing the influence of growing conditions and environmental factors on the chemical profiles of single cultivars are carried out.

## 1. Introduction

Olive (*Olea europaea* L.) belongs to the Oleaceae family and is a tree of significant biological, economic, and cultural importance. Its long record of cultivation led to the multiple phenotypic expressions, usually described as cultivars, each one characterized by specific morphometric and biological features [1]. It is becoming more and more evident that different cultivars are also characterized by the production of different specialized metabolites [2,3,4], and since these compounds are responsible for many properties of olives and olive oil, it is crucial to study this aspect.

The world olive germplasm is made up by more than 2600 cultivars, 600 of which are cultivated in Italy [5]. This high variability is certainly due to the fact that olive is an allogamous species characterized by a high level of hetero-pollination. However, they are also the result of a long history of selection and adaptation to specific environmental conditions. Endemic cultivars, in particular, evolved for a very long period in a specific local area and developed adaptative traits that are well-integrated with the environmental, agronomic, cultural, and traditional landscape features of the site [6]. This huge number of cultivars is an enormous resource in terms of biodiversity, but on the other hand, their taxonomic classification in not a trivial issue. There are several cases of synonyms and homonyms (mainly local names), generating a high degree of uncertainty [1]. Furthermore, there is a lack of a unified and effective classification system. The classical method to distinguish between cultivars is based on morphological and bio-agronomic parameters. Lately, this method is also supported by ultrastructural, biochemical, and molecular characters, including information on the specific chemical content often obtained through metabolomics [3,5,6,7].

The knowledge on the chemical content, especially concerning the specialized metabolites, is important not only for the purpose of cultivar classification but, also, to understand the added value that these metabolites confer to the cultivars producing them in terms of the agronomic features and nutraceutical potential. Indeed, each organ of olive trees produces and/or accumulates several specialized metabolites [8]. Concerning the drupes, they contain relatively large amounts of phenolic compounds. The main phenolics in olive leaves and drupes are compounds derived from 3,4-dihydroxyphenylethanol (hydroxytyrosol) and *p*-hydroxyphenylethanol (tyrosol) bound to secoiridoids like elenolic acid or its derivatives [9,10,11]. Glucosilated forms like oleuropein and olecanthal have also been reported, along with free tyrosol and hydroxytyrosol [9,10,11]. Caffeic acid and phenyletanoid glycosides are also components of the metabolome of olive leaves and drupes [9,10,11].

Phenolic compounds in the olive drupes are not only important as putative biomarkers of different cultivars but, also, for their potential use as nutraceuticals. Olives and olive oils can be considered as built-in functional foods [12], due to the presence of compounds endowed with specific biological activities. These compounds have a well-known antioxidant activity, and for oleuropein, this has also been shown in vivo—at least, in animal models [13,14,15]. Oleuropein seems to have also a cholesterol-lowering activity [15]. Moreover, the structurally related compound oleocanthal has been shown to possess anti-inflammatory activity by inhibiting cyclooxygenases [16,17].

In this study, 19 cultivars of olives (autochthonous and allochthonous) cultivated in Cilento, Vallo di Diano and Alburni National Park (PNCVDA, Italy) have been studied through a metabolomics-based approach. The territory of PNCVDA is characterized by an extensive cultivation of olives. Many autochthonous cultivars have been selected, because they are resistant to the specific pedoclimatic conditions. However, in recent years, the depletion and extensive replacement with new, more productive, varieties have resulted in a progressive reduction of varietal biodiversity with the alarming consequences of genetic erosion and loss of biodiversity. In this context, the knowledge of potential olive and olive oil biodiversity resources represents a crucial issue to preserve and to promote a socioeconomic development. A greater sustainability of modern agriculture is indeed required, especially in protected areas where the safeguard of old local varieties is a focal point for the conservation of traditional farming systems and landscapes [18].

In order to deepen the knowledge about autochthonous cultivars, which is doubtless the first step towards their protection, several studies have been carried out to study their distribution and to characterize them from the morphological, genetical, and phytochemical points of view [3,19]. The phytochemical analysis is critical, because on one hand, it could give information on specific biomarkers for different cultivars, and, on the other hand, it directly gives a clue about the nutraceutical value. In this framework, a metabolomics approach offers the advantages of a faster screening compared to the classical phytochemical study while also allowing the detection of compounds that were not previously reported.

Therefore, an NMR-based metabolomics study was carried out on 19 cultivars, allowing the identification of the most phenolic-rich cultivars, along with the identification of specific metabolites.

## 2. Results and Discussion

### 2.1. H-NMR Analysis

The 19 olive cultivars collected in PNCVDA (Table 1)were analyzed in terms of the metabolite contents of the pulps of the drupes.

#### 2.1.1. Multivariate Data Analysis: PCA on NMR Data

The ^1^H-NMR spectra obtained by the extraction of the plant material in a mixture of 1:1 CD_3_OD:phosphate buffer in D_2_O were processed, and bucketing was carried out. The integral table was analyzed via multivariate data analysis.

The PCA score plot (Figure 1) showed a distinction in two clusters according to PC1, i.e., a cluster described by the positive scores, and one characterized by the negative scores.

From the analysis of the loading plot (Figure 2A), it was possible to identify the variables that were correlated with PC1. It, indeed, showed a direct correlation with signals in the sugar region and an indirect correlation with signals in the region of the aromatic and aldehyde protons. Furthermore, an indirect correlation was also observed with signals in the regions 1.72–2.00 ppm and 2.68–3.12 ppm, along with a few other signals in the aliphatic region.

Within each cluster, a gradient was observed along PC2. This component showed an indirect correlation with signals in the aliphatic region and a direct correlation with signals in the aromatic region and a few aldehydic signals. For what concerns the aldehyde and aromatic signals, it was clear that those directly correlated with PC1 were indirectly correlated with PC2 and vice versa (Figure 2). Therefore, the metabolites generating the signals resonating in these regions might be important for the discrimination of the different cultivars. Furthermore, signals potentially belonging to olefinic protons also showed a direct correlation with PC2.

Although along PC2 there was no clear distinction in groups- but rather a gradient was observed- it is however possible to separate each cluster into two distinct groups according to their position on the PCA score plot. Overall, the samples could then be divided into four groups based on their positions in the different PCA quadrants.

Since PC1 is indirectly correlated with the aromatic and aldehydic signals and PC2 is directly correlated with the same signals, it is already possible to state that the samples located in the right lower quadrant of the PCA (Group I, Figure 1) were characterized by the presence of smaller amounts of aromatic compounds compared to the other cultivars. Ricippudda (RI) only showed traces of these signals and is the cultivar with the lowest amount, but also Salella (SA), Rotondella (RT), and Nostrale (NO) were not particularly rich in these signals (Figure 3).

All of the other cultivars were characterized by a very rich aromatic region.

The samples in the upper-right quadrant of the PCA score plot (GroupII, Figure 1) were characterized by signals in the aromatic and olefinic regions, but most notably by the presence of metabolites characterized by an aldehydic signal at δ_H_ 9.09. Although in the cultivars that were closer to the zero value of the PC2 axis other signals in the aldehydic region were also detected, in the Marinella samples (MA) δ_H_ 9.09 was still the predominant one (Figure 4).

The cultivars Marinella (MA), Provenzale (PR), Sanginara (SN), Oliva amara (AM), and Cammarotana (CA) constituted Group II and were characterized by the presence of this signal at δ_H_ 9.09 and further signals in the aromatic region.

Since the signal at δ_H_ 9.09 was directly correlated with PC2, we also expected it to be in a significant intensity in the samples of Racioppella (RO) and Guglia (GU), located in the upper-left quadrant of the PCA score plot (Group III, Figure 1). However, in these spectra, signals indirectly correlated with PC1 were also detected in the aromatic region—in particular, doublets in the regions 6.00–6.50 ppm and 7.00–7.50 ppm (Figure 5). In the samples closer to the zero value of the PC2 axis (PI, FT, and GR), several signals of other aldehydic protons were also detected.

Finally, the samples present in the lower-left quadrant (Group IV, Figure 1) showed once again signals of aromatic and aldehydic protons, but characterized by different chemical shifts and multiplicity compared to the other cultivars (Figure 6). Furthermore, these metabolites were less abundant in the cultivars of Group IV compared to the cultivars of Groups II and III.

Although, in all of the four groups of cultivars herewith described, the signals in the aliphatic region appeared to change, most of them did change in accordance with the aromatic, olefinic, and aldehydic signals. Therefore, they might arise from protons belonging to the same molecules. In order to prove this and to identify the chemicals responsible for the observed separation, 2D NMR analyses were carried out.

#### 2.1.2. Identification of the Metabolites in the Extracts

The PCA analysis showed that the compounds generating the aldehydic and some aromatic and olefinic signals might be important for the cultivar discrimination. Therefore, 2D NMR analyses (Appendix A) were carried out on selected samples with the aim of identifying the compounds generating these signals in the NMR spectra.

Among the cultivars of Group II, Cammarotana samples were useful for the identification of two of the compounds detected in the extracts. Indeed, two aldehydic signals were detected: one at δ_H_ 9.24 and another one at δ_H_ 9.09. Based on the 2D NMR data, it was possible to assign the proton at δ_H_ 9.24 to compound **1** (Figure 7); the HMBC experiment showed the correlations reported in Table 2. A TOCSY experiment was useful to confirm the spin systems belonging to this compound. From the 2D NMR of this extract, it was also possible to identify the dihydroxyphenylethanolelenolic acid dialdehyde (DHPEA-EDA, **2**). The presence of the aromatic signals as two doublets and a double of doublets, respectively, at δ_H_ 6.70, 6.79, and 6.58, the benzylic proton at δ_H_ 2.77, and the diagnostic signals of the iridoid moiety at δ_H_ 9.09, 6.72, and 1.90, along with the correlation reported in Table 2, allowed us to identify this compound in the extract. The data were in accordance with the literature [20].

In the spectra of the Group III, aldehyde signals not yet identified were present, especially in Pizzulella (PI). However, since the same signals were also detected, along with other related ones, in the Pisciottana (PS) cultivar, 2D NMR analyses were carried out on the latter, allowing the identification of several metabolites at the same time. Signals resonating at δ_H_ 9.04 (doublet) and a double of doublets at δ_H_ 9.00 were detected and observed to always be at a 1:1 ratio. Based on the HSQC and HMBC data (Table 2), it was possible to identify the dialdehydic form of oleuropein, also known as oleomissional [21] (**3**, Figure 7). Finally, another aldehydic compound detected in the extracts was oleocanthal (**4**). Also in this case, their identity was confirmed thanks to 2D NMR data (Table 2), which were in accordance with those reported in the literature [22].

Oleuropein (**5**), demethyloleuropein (**6**), nüzhenide (**7**), and the hydrated form of the aglycone of oleuropein (**8**) (Figure 8) were present only in a few cultivars and in low amounts. Their identity was therefore putatively established thanks to the comparison with the previously published data [3,20,23]. The presence of oleuropein was suggested by the singlets at δ_H_ 5.85 (H-1) and 7.54 (H-3), the quartet at δ_H_ 6.06 (H-8), the methyl at δ_H_ 1.58 (H-10), and the anomeric proton of glucose at δ_H_ 4.60. Furthermore, signals of the hydroxytyrosol moiety were detected. Diagnostics signals of demethyloleuropein were the singlets at δ_H_ 5.76 (H-1) and 7.27 (H-3) and the quartet at δ_H_ 6.00 (H-8). The presence of nüzhenide was, on the other hand, suggested by the singlets at δ_H_ 5.96 (H-1) and 7.55 (H-3), the quartet at δ_H_ 6.10 (H-8), and the methyl at δ_H_ 1.78 (H-10). Finally, the hydrated form of the aglycone of oleuropein was detected thanks to the olefinic proton at δ_H_ 7.59 (H-3), the methine at δ_H_ 3.26 (H-5), the signal at δ_H_ 4.86 (H-1), and the methyl at δ_H_ 1.28 (H-10). This compound is present as different stereoisomers, but it is not possible to discriminate them in the mixture.

Besides the signals of the hydroxytyrosol and tyrosol secoiridoid derivatives, signals belonging to cornoside and to halleridone [24] (**9** and **10**, Figure 8) were also detected and identified based on the correlations observed in the Grossale cultivar (Table 2). Finally, the Racioppa extract allowed us to attribute signals belonging to verbascoside [20,25] (**11**, Table 2 and Figure 8).

### 2.2. Classifications of the Analyzed Cultivars Based on Their Metabolite Content

Although several other metabolites are present in the extracts, from the previous analyses, it was clear that the discrimination between the different cultivars was possible thanks to the compounds generating aromatic, olefinic, and aldehydic signals. The signals in this region were therefore integrated and normalized by the integral of the internal standard TMSP. The cultivars are listed in Table 3 according to their decreasing content for these metabolites (RO was the cultivar with the highest value for the integral of the region between 5.5 and 9.5 ppm). Based on diagnostic signals of the identified metabolites, it was also possible to tell which metabolites are present in each cultivar (Table 3).

## 3. Discussion

The metabolomics analysis of the drupes of 19 autochthonous olive cultivars endemic of the PNCVDA allowed us to classify them based on their richness in potential nutraceutical compounds, both from a quantitative and a qualitative point of view. Focusing on the variables identified through PCA analysis of the NMR data (Figure 1 and Figure 2) and after the attribution of these variables to metabolites thanks to the 2D NMR analyses of selected spectra (Table 3, Appendix A), it was possible to classify the studied cultivars based on the richness in these metabolites. Furthermore, it was possible to define the composition, relative to the metabolites of interest, of each cultivar.

The cultivars Racioppella, Guglia, Pizzulella, Oliva amara, and Racioppa were the ones characterized by the highest content in specialized metabolites (Table 3). The main metabolite in these cultivars was DHPEA-EDA (Figure 4, Figure 5, Figure 6, Figure 7 and Figure 8). However, they were all characterized by the presence of several different metabolites (Table 3). In the cultivars Pizzulella and Racioppa, also signals of oleacein and verbascoside, respectively, stood out.

The cultivars characterized by the lowest concentration of the target compounds were Nostrale, Salella, Rotondella, and Ricippudda (Figure 3). Among these, however, Nostrale had a more diversified profile, since signals from many different compounds were detected (Table 3).

Among the detected compounds, some (**1**–**4**) seemed to be rather common in the analysed cultivars, while the less common cornoside and halleridone were only detected in specific cultivars (Table 3). Oleuropein and its derivatives seemed to have a nonuniform distributions as well. Finally, verbascoside was also only detected in some of the cultivars. Could any of these compounds be considered as a potential biomarker? None of the compounds were typical for only one cultivar; however, the specific profile of the cultivar could be taken into consideration to establish a sort of fingerprinting approach for cultivar identification. At this point, it is important to highlight that the content in specialized metabolites of any plant material could be affected by several variables [20,26,27]. In this study, one of the most important was excluded thanks to the specific approach for sample collection: all of the drupes were collected at the same ripening stage. However, many other factors that could impact the olive metabolome will need to be investigated, particularly altitude, rainfall, and temperature. It has been reported, for example, that fruits harvested at higher altitudes are characterized by higher tocopherol and total phenolic contents [28].

The present study only shows that, for most of the cultivars, a specific and unique profile was detected. A crucial next step will be to address the variability in the metabolome of each cultivar based on the cultivation site and environmental conditions. Although previous studies have been reported in which either the influence of the cultivar or the influence of the environment on the content in specialized metabolites have been explored [2,3,29], metabolomics now makes it possible to design studies in which the combined influences of the different factors could be addressed. This information is crucial to assess the nutraceutical value of each cultivar because the latter is directly correlated to the content in some phytochemicals.

Based on the data herewith discussed, and on the current knowledge, the cultivars Racioppella, Guglia, Pizzulella, Oliva amara, and Racioppa are those potentially characterized by the highest nutraceutical value. Among these cultivars, Racioppella, Guglia, Pizzulella, and Racioppa were particularly interesting for the presence of oleocanthal. This compound has a well-proven anti-inflammatory activity [16,17], shared also with the verbascoside [30]. Phenolic compounds, in general, have been largely studied for their antioxidant and radical scavenging properties. Oleuropein and oleocanthal, in particular, have not only been tested in vitro but, also, in vivo or at least in in vivo models [31]. The olive drupes herewith analyzed contained, however, relatively low amounts of oleuropein that seems to be abundant in leaves [32]. These compounds, along with the other secoiridoid derivatives described, are also characterized by documented antimicrobial activity against several pathogens [33] and potential anticancer activity [32]. It has been shown, however, that phytochemical mixtures might have different activity than the pure compounds due to synergistic, additive, but, also, antagonistic effects.

Further studies are therefore needed to experimentally assess the nutraceutical potential of the analyzed cultivars. This has to go hand-in-hand with the evaluation of the variability of the content of these metabolites in the cultivars of interest depending on different factors.

## 4. Materials and Methods

### 4.1. Sample Collection

In this study, 19 olive cultivars (Table 1) were collected in the PNCVDA (Table 1) and analyzed. In particular, the metabolite content of the pulps of the drupes was explored by an NMR-based metabolomics approach.

Since the ripening stage strongly affects the contents in metabolites, it was important to choose the right moment for the sample collection. It was possible to assess the ripening stage based on a previously reported method and based on weekly monitoring on the sugar content of the drupes [33,34]. Furthermore, at the harvesting time, attention was paid to the phytosanitary state of the plants and of the drupes, in particular, selecting those that did not present any attack of pathogens.

For each cultivar, samples were collected (3 biological replicates constituted by a 50-mL falcon tube -Falcon, Berlin, Germany- filled with the drupes). Samples were immediately frozen in liquid nitrogen and freeze-dried as soon as they were transported to the laboratory. The pulps of the drupes were then separated from the seeds and freeze-dried again. The samples were then ground in liquid nitrogen and stored at −80° until further analyses.

### 4.2. Metabolomics Analysis

#### 4.2.1. Extraction

Freeze-dried and powdered plant material (50 mg) was transferred to a 2-mL Eppendorf tube (Eppendorf, Montesson, France). The NMR samples were prepared by mixing the lyophilized plant material with 1.5 mL of the NMR solvent consisting of a phosphate buffer (90 mM; pH 6.0) in D_2_O (Sigma-Aldrich, St. Louis, MO, USA) containing 0.01% *w/v* trimethylsilylpropionic-2,2,3,3-*d*_4_ acid sodium salt (TMSP; Sigma-Aldrich) and CD_3_OD (Sigma-Aldrich) (1:1). The mixture was vortexed at room temperature for 1 min, ultrasonicated (Bandelin Sonorex-Berlin, Germany, RX100) for 20 min, and centrifuged (Eppendorf 5415R F45-24-11 rotor) at 13,000 rpm for 10 min. An aliquot of 600 mL of the supernatant was transferred to an NMR tube.

#### 4.2.2. NMR Analysis

NMR spectra were recorded at 25 °C on a Varian (Palo Alto, CA, USA) Mercury Plus 300 Fourier transform NMR operating at 300.03 MHz for ^1^H and 75.45 MHz for ^13^C. CD_3_OD was used as the internal lock. One-dimensional and 2D NMR spectra were acquired using Varian standard pulse sequences and as previously described [20].

Two-dimensional NMR analyses were carried out on selected samples using standard library sequences for HSQC, HMBC, COSY, and TOCSY.

#### 4.2.3. Multivariate Data Analysis

^1^H NMR spectra were scaled to total intensity and bucketed, reducing them to integral segments with a width of 0.04 ppm with ACDLABS 12.0 ^1^H NMR processor (ACDLABS, Toronto, ON, Canada). The regions at δ −0.02–0.02, 4.70–5.00, and 3.30–3.34 were excluded from the analysis (by indicating them as dark regions before integration) because of the residual TMSP and solvents signals. Principal component analysis (PCA) was performed with the SIMCA-P software (version 14.0, Umetrics, Umeå, Sweden) with scaling based on Pareto.

## 5. Conclusions

The present study, aimed at determining the phytochemical profiles of the drupes of different olive cultivars of the PNCVDA, showed specific and unique profiles for most of the analyzed cultivars. Whether or not this information can be used for the classification of cultivars needs to be assessed, with further studies aimed at determining the variability of the metabolome for each of the analyzed cultivar depending on several factors. Nevertheless, the knowledge on the specialized metabolites present in each cultivar is essential to assess their nutraceutical potential. The present data showed that the cultivars Racioppella, Guglia, Pizzulella, and Racioppa were particularly interesting for the presence of oleocanthal, a molecule with several proven biological activities, although several other cultivars were characterized by the presence of potential health-promoting compounds.

## Figures and Tables

**Figure 1 molecules-26-03845-f001:**
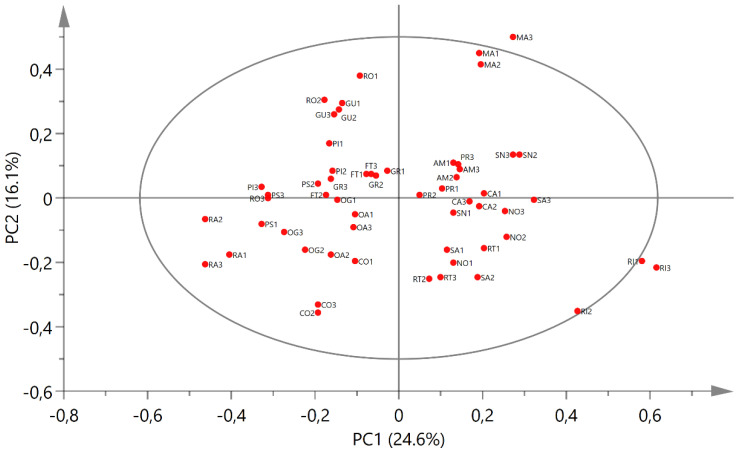
Principal component analysis of ^1^H NMR data: score scatter plot of PC1 versus PC2. The ellipse represents the Hotelling T2 with 95% confidence. The abbreviations used are reported in Table 1. Numbers 1–3 indicate three independent biological replicates for each cultivar.

**Figure 2 molecules-26-03845-f002:**
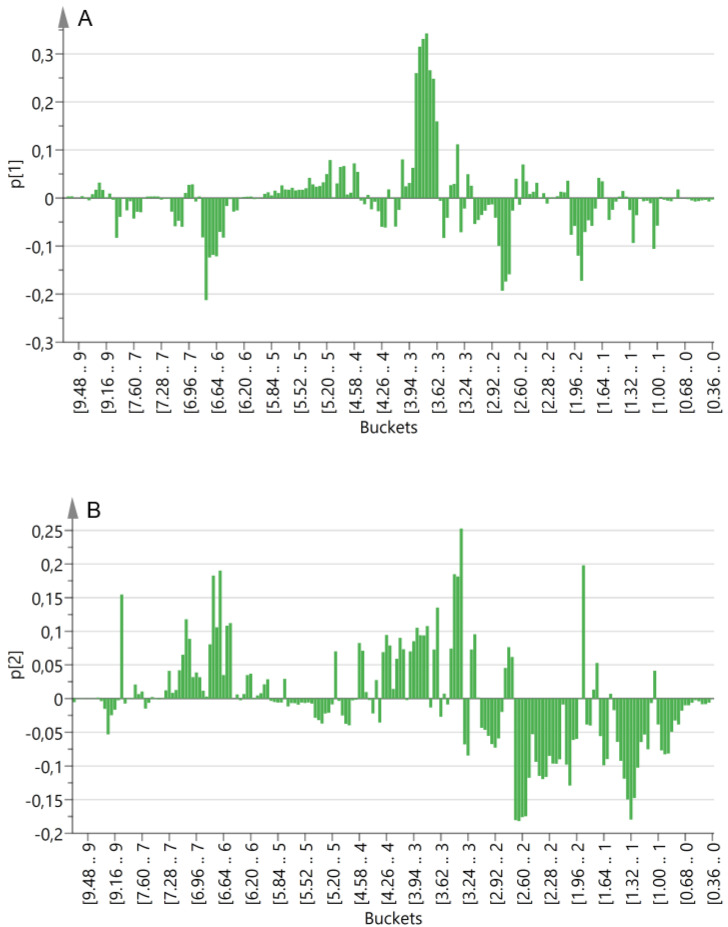
PCA loading plots: (**A**) PC1 and (**B**) PC2.

**Figure 3 molecules-26-03845-f003:**
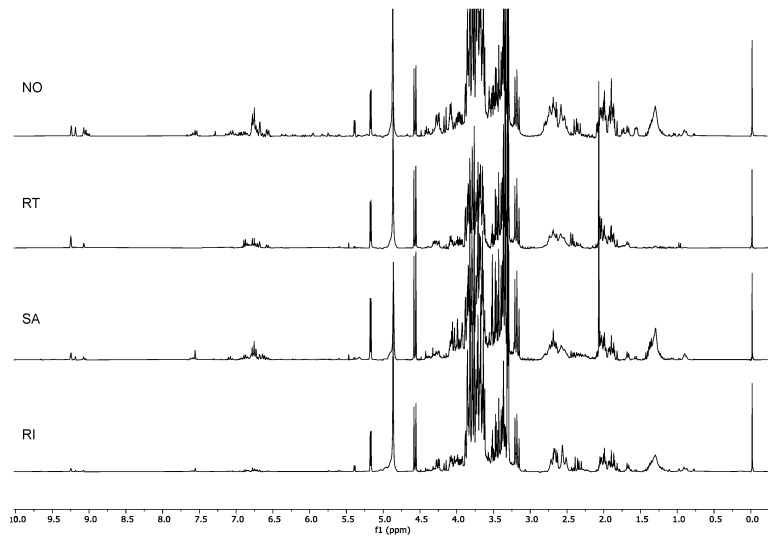
^1^H-NMR spectra of Group I: Ricippudda (RI), Salella (SA), Rotondella (RT), and Nostrale (NO).

**Figure 4 molecules-26-03845-f004:**
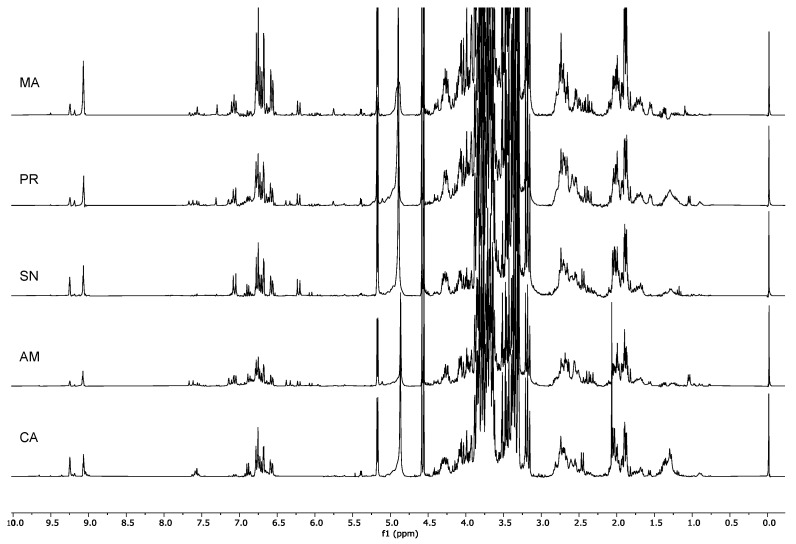
^1^H-NMR spectra of Group II: Cammarotana (CA); Oliva amara (AM), Sanginara (SN), Provenzale (PR), and Marinella (MA).

**Figure 5 molecules-26-03845-f005:**
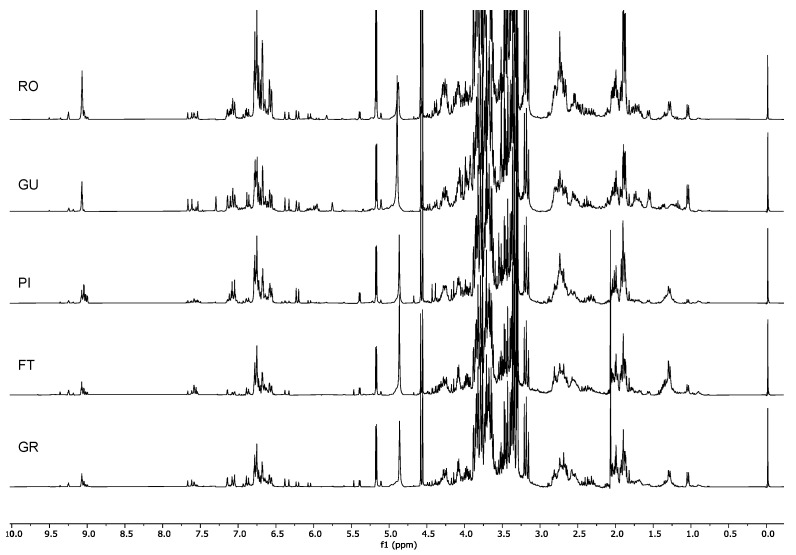
^1^H-NMR spectra of Group III: Grossale (GR), Femminella (FT), Pizzulella (PI), Guglia (GU), and Racioppella (RO).

**Figure 6 molecules-26-03845-f006:**
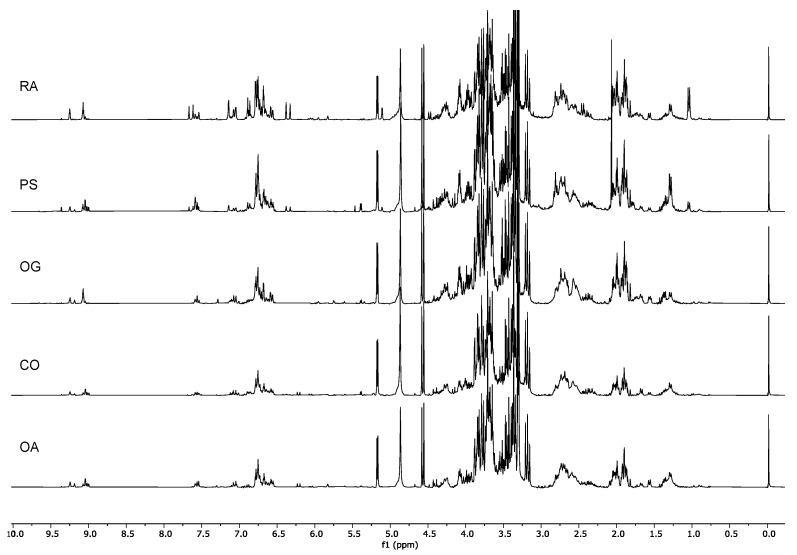
^1^H-NMR spectra of Group IV: Ogliara (OA), Carolea (CO), Ogliastro (OG), Pisciottana (PS), and Racioppa (RA).

**Figure 7 molecules-26-03845-f007:**
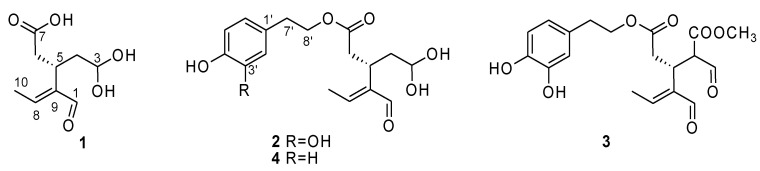
Compounds **1**–**4** identified in the extracts and responsible for the differentiation of the samples.

**Figure 8 molecules-26-03845-f008:**
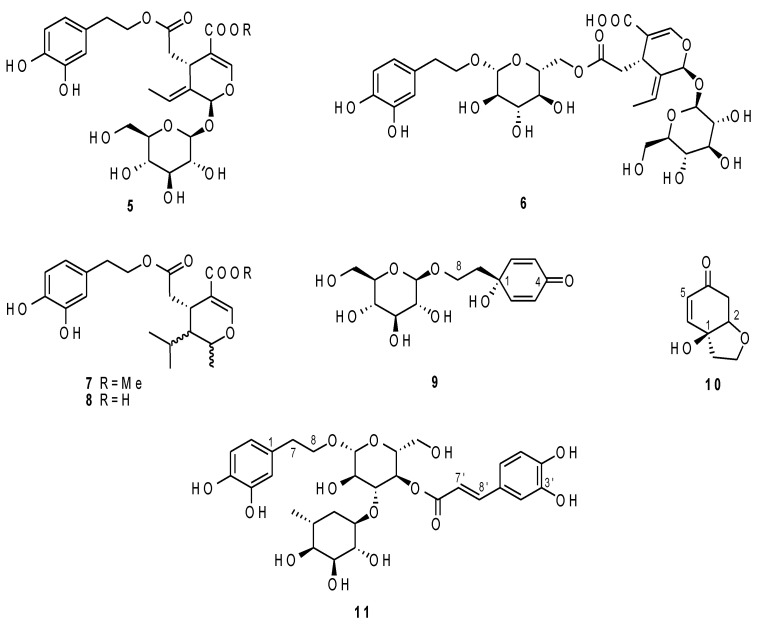
Compounds **5**–**11** identified in the extracts and responsible for the differentiation of the samples.

**Table 1 molecules-26-03845-t001:** List of the studied cultivars, abbreviations, and collection sites.

Cultivar	Abbreviation	Collection Site
Oliva amara	OA	Morigerati
Cammarotana	CA	Orria
Carolea	CO	Vallo della Lucania
Femminella	FT	Torraca
Grossale	GR	Torraca
Guglia	GU	Morigerati
Marinella	MA	Morigerati
Nostrale	NO	Castel S. Lorenzo
Ogliara	OA	Agropoli
Ogliastro	OG	Ogliastro
Pizzulella	PI	Orria
Provenzale	PR	Torre Orsaia
Pisciottana	PS	Pisiotta
Racioppa	RA	Vallo della Lucania
Ricippudda	RI	Caselle in Pittari
Racioppella	RO	Vallo della Lucania
Rotondella	RT	Giungano
Salella	SA	Salento
Sanginara	SN	Stella Cilento

**Table 2 molecules-26-03845-t002:** NMR data of the compounds identified in the extracts acquired at 300 MHz in 1:1 CD_3_OD: phosphate buffer in D_2_O.

	Position	^1^H	^13^C	HMBC Correlations
Compound **1**	1	9.24	198.2	C5, C9
	3	4.36	99.8	C5
	4	1.72/2.05	37.5	C3, C5, C6, C9
	5	3.26	30.2	C1, C3, C4, C8, C9
	6	2.54/2.68	37.5	C4, C5, C7, C9
	7		178.9	
	8	6.92	157.3	C1, C5, C9, C10
	9		146.3	
	10	2.05	14.8	C1, C8, C9
DHPEA-EDA (**2**)	1	9.09	200.6	C5, C8, C9
	3	4.33	99.8	
	4	1.72/2.04	37.4	C3, C5, C6, C9
	5	3.18	32.5	C1, C3, C4, C8, C9
	6	2.54/2.72	38.4	C4, C5, C7, C9
	7		174.3	
	8	6.72	160.7	C1, C5, C9, C10
	9		145.1	
	10	1.90	17.3	C1, C8, C9
	1′		130.6	
	2′	6.70	116.1	C1′, C6′, C7′
	3′		144.3	
	4′		143.6	
	5′	6.79	115.9	C1′, C3′, C4′, C6′
	6′	6.58	120.7	C1′, C2′, C5′, C7′
	7′	2.77/2.80	33.6	C7
	8′	4.17/4.28	65.6	C1′, C7, C7′
Oleomissional (**3**)	1	9.04	197.5	C4, C8, C9
	3	9.00	197.1	C4, C9
	4	3.47	32.5	C1, C3, C7, C8, C9
	7		173.5	
	8	6.78	157.9	C1, C5, C9, C10
	9		140.9	
	10	1.90	15.5	C1, C8, C9
	1′		130.6	
	2′	6.70	116.1	C1′, C6′, C7′
	3′		144.3	
	4′		143.6	
	5′	6.79	115.9	C1′, C3′, C4′, C6′
	6′	6.58	120.7	C1′, C2′, C5′, C7′
	7′	2.75/2.80	33.6	C7
	8′	4.17/4.28	65.6	C1′, C7, C7′
Oleocanthal (**4**)	1	9.10	198.4	C5, C8, C9
	3	4.28	96.9	
	4	1.77/2.01	38.5	C3, C5, C6, C9
	5	3.2	30.3	C1, C3, C4, C8, C9
	6	2.56/2.75	37.5	C4, C5, C7, C9
	7	-	175.3	
	8	6.76	157	C1, C5, C9, C10
	9	-	142.9	
	10	1.91	14.7	C1, C8, C9
	1′		130.6	
	2′/6′	7.05	129.7	C1′, C6′, C7′
	3′/5′	6.72	115.7	
	4′	-	154.6	
	7′	2.76	33.9	
	8′	4.09/4.22	65.6	C1′, C7, C7′
Cornoside (**9**)	1		70.8	
	2/6	7.06	156.2	C2, C3, C4, C6
	3/5	6.23	129.5	C1, C2, C4, C5
	4		189.8	
Halleridone (**10**)	1		77.5	
	2		83.0	
	3		39.0	
	4	6.08	199.9	
	5	6.93	130.4	C1, C3, C4
	6		157.6	C2, C4, C5
Verbascoside (**11**)	1′		125.1	
	2′	7.12	113.3	C3′, C4′, C6′
	3′		148.4	
	4′		145.2	
	5′	6.89	112.3	C1′, C3′, C4′
	6′	7.08	121.4	C2′, C4′
	7′	7.67	147.6	C2′. C3′, C6′, C9′
	8′	6.36	113.9	C1′, C9′
	Glc-1	4.49	102.3	
	Rha-1	5.13	101.8	
	Rha-6	1.06	17.3	

^1^ 2D NMR spectra are reported in the Appendix A. The carbon signals were deduced either from HSQC (the ones bound to protons) or from HMBC. In some cases, COSY and TOCSY experiments were used to support the characterization.

**Table 3 molecules-26-03845-t003:** Phytochemical compositions of the analyzed cultivars.

	Metabolites										
Cultivar	1	2	3	4	5	6	7	8	9	10	11
RO	●	●	●	●	●			●		●	●
GU	●	●		●		●	●			●	●
PI	●	●	●	●				●		●	●
MA	●	●		●		●			●	●	
RA	●	●	●	●	●		●	●			●
PS	●	●	●	●	●			●			●
PR	●	●		●		●			●	●	●
GR	●	●	●	●				●		●	●
FT	●	●	●	●				●			●
AM	●	●		●					●	●	●
OG	●	●	●	●		●	●				
SN	●	●		●					●	●	
CA	●	●									
OA	●	●		●	●			●			
CO	●	●	●	●				●			
NO	●	●	●	●	●	●	●	●	●		●
SA	●	●	●	●							
RT	●	●	●								
RI	●	●									

## Data Availability

The raw data supporting the conclusions of this article will be made available by the authors upon request, without undue reservation.

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
