# Peer review of "Phytochemical Characterization of Olea europaea L. Cultivars of Cilento National Park (South Italy) through NMR-Based Metabolomics"

_molecules, 2021, doi:10.3390/molecules26133845_

Round 1

Reviewer 1 Report

This work is interesting. The experiences were well outlined and well carried out. In my opinion it should be accepted.

Author Response

This work is interesting. The experiences were well outlined and well carried out. In my opinion it should be accepted.

R: Thank you for the endorsement

Reviewer 2 Report

1 Introduction part needs to be condensed.

2 The chemical shift on the left side of the H NMR spectrum should be adjusted to 10.0 in Fig.3-6.

3 The format of the references is inconsistent. For example, the first letter of the word in the title of the cited article has lowercase or uppercase; the journals have the full spelling or the abbreviated format, please check one by one.

Author Response

1 Introduction part needs to be condensed.

R: We condensed whenever possible

2 The chemical shift on the left side of the H NMR spectrum should be adjusted to 10.0 in Fig.3-6.

R: Thank you for noticing. This has been changes.

3 The format of the references is inconsistent. For example, the first letter of the word in the title of the cited article has lowercase or uppercase; the journals have the full spelling or the abbreviated format, please check one by one.

R: The reference list has been reviewed accordingly.

Reviewer 3 Report

This manuscript describes phytochemical characterization of olea europaea L. cultivars from Cilento National Park in southern Italy using 1D, 2D NMR. The use of NMR based metabolomics study allowed the identification of the most phenolic-rich cultivars among the 19 cultivars studied. I think the metabolomics method is advantageous in that it offers faster screening and will be appealing to the scientist in the field and broad readership of Molecules. I recommend publishing after some minor revisions.

  1. Figure 7 was mentioned on page 7 of the manuscript but the actual figure 7 appeared in page 11. I recommend to separate Figure 7 into two figures with compound 1-4 in one figure and insert this figure before table 2 so that the readers don’t have to go 4 pages later to look for the structures of compounds 1-4.
  2. There are some concluding comments in the manuscript. But it is not very clear what are the conclusions of this manuscript without a separate section of “Conclusion”. It is better to have a separate section 5 titled “Conclusion” after section 4. “Materials and Methods” at the end of the manuscript to highlight the conclusion of this research.

Author Response

This manuscript describes phytochemical characterization of olea europaea L. cultivars from Cilento National Park in southern Italy using 1D, 2D NMR. The use of NMR based metabolomics study allowed the identification of the most phenolic-rich cultivars among the 19 cultivars studied. I think the metabolomics method is advantageous in that it offers faster screening and will be appealing to the scientist in the field and broad readership of Molecules. I recommend publishing after some minor revisions.

    Figure 7 was mentioned on page 7 of the manuscript but the actual figure 7 appeared in page 11. I recommend to separate Figure 7 into two figures with compound 1-4 in one figure and insert this figure before table 2 so that the readers don’t have to go 4 pages later to look for the structures of compounds 1-4.

R: Thank you for the suggestion. The figure is now split in two.

    There are some concluding comments in the manuscript. But it is not very clear what are the conclusions of this manuscript without a separate section of “Conclusion”. It is better to have a separate section 5 titled “Conclusion” after section 4. “Materials and Methods” at the end of the manuscript to highlight the conclusion of this research.

R: The conclusion section has been added